# *One Health* Surveillance of Antimicrobial Resistance Phenotypes in Selected Communities in Thailand

**DOI:** 10.3390/antibiotics11050556

**Published:** 2022-04-21

**Authors:** Duangdao Sudatip, Surapee Tiengrim, Kittipong Chasiri, Anamika Kritiyakan, Wantanee Phanprasit, Serge Morand, Visanu Thamlikitkul

**Affiliations:** 1Faculty of Public Health, Ubon Ratchathani Rajabhat University, Ubon Ratchathani 34000, Thailand; duangdao1502@gmail.com; 2Faculty of Public Health, Mahidol University, Bangkok 10400, Thailand; wantanee.pha@mahidol.ac.th; 3Faculty of Medical Technology, Mahidol University, Nakhon Pathom 73170, Thailand; surapee.tie@mahidol.ac.th; 4Faculty of Tropical Medicine, Mahidol University, Bangkok 10400, Thailand; kittipong.cha@mahidol.ac.th (K.C.); serge.morand@cirad.fr (S.M.); 5Faculty of Veterinary Technology, Kasetsart University, Bangkok 10220, Thailand; anamikanose.k@gmail.com; 6Faculty of Medicine Siriraj Hospital, Mahidol University, Bangkok 10700, Thailand

**Keywords:** antimicrobial resistance phenotype, *One Health* surveillance, community, Thailand

## Abstract

Integrated surveillance of antimicrobial resistance (AMR) using the *One Health* approach that includes humans, animals, food, and the environment has been recommended by responsible international organizations. The objective of this study was to determine the prevalence of AMR phenotypes in *Escherichia coli* and *Klebsiella* species isolated from humans, pigs, chickens, and wild rodents in five communities in northern Thailand. Rectal swabs from 269 pigs and 318 chickens; intestinal contents of 196 wild rodents; and stool samples from 69 pig farmers, 155 chicken farmers, and 61 non-farmers were cultured for *E. coli* and *Klebsiella* species, which were then tested for resistance to ceftriaxone, colistin, and meropenem. The prevalence of ceftriaxone-resistant *E. coli* and *Klebsiella* species in pigs, chickens, rodents, pig farmers, chicken farmers, and non-farmers was 64.3%, 12.9%, 4.1%, 55.1%, 38.7%, and 36.1%, respectively. Colistin resistance in pigs, chickens, rodents, pig farmers, chicken farmers, and non-farmers was 41.3%, 9.8%, 4.6%, 34.8%, 31.6%, and 24.6%, respectively. Meropenem resistance was not detected. The observed high prevalence of AMR, especially colistin resistance, in study food animals/humans is worrisome. Further studies to identify factors that contribute to AMR, strengthened reinforcement of existing regulations on antimicrobial use, and more appropriate interventions to minimize AMR in communities are urgently needed.

## 1. Introduction

Antimicrobial resistance (AMR) is a major evolving global health problem that is associated with high morbidity, high mortality, and substantial economic loss [1,2]. Several studies of AMR burden in humans conducted in Thailand found and reported enormous AMR-related health and economic burdens [3,4,5,6].

The World Health Organization (WHO) in collaboration with the World Organisation for Animal Health (OIE) and the Food and Agriculture Organization of the United Nations (FAO) endorsed and launched in 2015 a global action to combat AMR [7]. The foundational hypothesis of this action plan was that AMR affects sectors beyond human health, including animal health, agriculture, food security, and economic development. In response, a strategy called *One Health* was developed that includes all affected sectors and disciplines and that aims to reduce the prevalence of AMR via an integrated and unified approach among stakeholders, with the ultimate aim of sustainably balancing and optimizing the health of people, food animals, and ecosystems. One of the five strategic objectives of this action plan is to strengthen the knowledge and evidence base via surveillance and research. Particularly important gaps in knowledge that need to be filled include information specific to the incidence, the prevalence across pathogens, and the geographical patterns of AMR; understanding how resistance develops and spreads; understanding how resistance circulates within and between humans and food animals, and through food, water, and the environment; the ability to rapidly characterize newly emerged resistance in microorganisms and elucidate the underlying mechanisms; and understanding the social science and behavior of antibiotic use in all sectors that are in any way responsible for any aspect of antibiotic use.

Integrated surveillance of AMR using the *One Health* approach, which includes humans, animals, food, and the environment, has been recommended by the WHO, OIE, and FAO [8]. By way of example, a target of AMR surveillance is the monitoring of the prevalence of extended-spectrum beta-lactamase (ESBL)-producing *Escherichia coli* across the human, food animal, and environmental sectors. Importantly, the AMR global action plan allows countries to modify their integrated AMR surveillance to include other cross-cutting pathogens and other resistance mechanisms, expand implementation of the program to different cities and provinces in that country to obtain added information/evidence regarding the spread of AMR in different sectors, and facilitate the implementation of holistic interventions to contain AMR.

The aim of this study was to determine the prevalence of AMR phenotypes for common or important antimicrobial agents in *E. coli* and *Klebsiella* species isolated from humans, food animals, and wild rodents living in/raised in/harvested from the same community among five selected study communities located in a province in northern Thailand.

## 2. Materials and Methods

The protocol for this study was approved by the Ethics Committee of the Faculty of Tropical Medicine, Mahidol University, Bangkok, Thailand, for human study (COA no. MUTM-2018-035-01) and by the Scientific Research Committee of Kasetsart University, Bangkok, Thailand, for animal study (COA no. ACKU 62-VTN-010).

### 2.1. Study Site and Duration

The study was conducted during 2018 and 2019 in five districts of a province located in the northern region of Thailand.

### 2.2. Study Population

#### 2.2.1. Animals

The pigs and chickens included in the study were raised on 1 of the 127 different privately owned farms (77 chicken farms and 50 pig farms) that the farm owners agreed to participate in the study. The largest pig farm included had 552 pigs and the largest chicken farm had 950 chickens. The researchers randomly selected 4 to 10 adult pigs per pig farm and 3 to 10 adult chickens per chicken farm for a total of at least 260 pigs and 280 chickens from all pig farms and all chicken farms, respectively. The study protocol estimated that at least 100 wild rodents living on or closely around the study farms would be trapped, sacrificed, and analyzed in this study.

#### 2.2.2. Humans

The included pig farmers and chicken farmers were aged 16–70 years, and they raised chickens and/or pigs full-time or part-time. These farmers worked at the pig or chicken farms that were included in our study, and they resided in 1 of the 5 study communities. This study randomly selected 1 to 2 pig farmers per pig farm and 1 to 2 chicken farmers per chicken farm for an estimated total of at least 60 enrolled pig farmers and 80 enrolled chicken farmers. At least 60 non-farmers aged 16–70 years who had no contact with farm animals but who lived in the same study communities as the farmers were also included in the study. Written informed consent to participate in the study and to have their stool samples collected was obtained from all human subjects.

### 2.3. Collection of the Study Samples

A stool sample from each pig and chicken was collected via rectal swab, and the swab was maintained in a Cary-Blair transport medium tube (BOENMED^®^ Boen Healthcare Co., Ltd.; Suzhou, China). All trapped wild rodents were sacrificed, after which intestinal content of each wild rodent was swabbed and the intestinal content swab was put into a Cary-Blair transport medium tube. Stool samples collected from farmers and non-farmers were stored in small plastic containers without preservatives. Rectal swab samples of pigs and chickens and intestinal content swab samples of wild rodents were kept at room temperature whereas stool samples of humans were kept in box containing ice. The collected samples were transported to the microbiology laboratory of the Division of Infectious Diseases and Tropical Medicine of the Department of Medicine, Faculty of Medicine, Siriraj Hospital, Mahidol University, Bangkok, Thailand, within 3 days of collection.

### 2.4. Microbiological Study of the Collected Samples

The target bacteria in this study were *E. coli* and *Klebsiella* species. Each rectal swab collected from pigs and chicken, and the intestinal content swab collected from wild rodent, was inoculated in 5 mL of tryptic soy broth (TSB) (Sigma-Aldrich Corporation, St. Louis, MO, USA) and incubated at 35 °C for 16–24 h or overnight. After incubation, 100 μL of each TSB culture was inoculated onto a MacConkey agar plate supplemented with ceftriaxone (2 μg/mL) and onto a MacConkey agar plate supplemented with colistin (1 μg/mL) to detect antibiotic resistant Gram-negative bacteria. Each stool sample collected from humans was taken in the amount of an inoculation loop (approximately 10 μL) and streaked on the aforementioned antibiotic-supplemented agar plates. Suspected colonies of Enterobacterales (lactose fermenter or pink colony) grown on these agar plates were subjected to manual biochemical tests, including the triple sugar iron test, lysine iron agar slants test, indole test, motility test, ornithine decarboxylase test, urease test, and malonate test, which were locally prepared in the laboratory using the purchased reagents/materials (Oxoid Ltd.; Hampshire, UK or BBL/Difco Diagnostic, Becton Dickinson; Sparks, MD, USA), and the oxidase test (BBL/Difco Diagnostic, Becton Dickinson; Sparks, MD, USA), to identify *E. coli* and *Klebsiella* species.

### 2.5. Antimicrobial Susceptibility Test of E. coli and Klebsiella Species Isolates

The minimum inhibitory concentration (MIC) of ceftriaxone, colistin, and meropenem against *E. coli* and *Klebsiella* species isolates was determined by the agar dilution method according to the Clinical Laboratory Standards Institute (CLSI) guidelines, and an MIC of ceftriaxone, colistin, and meropenem of ≥4 μg/mL was considered resistance to these three drugs [9]. *Escherichia coli* ATCC25922 was used as a control strain.

### 2.6. Data Analysis

Descriptive statistics were used to analyze and describe the data. The data, all of which were categorical, were compared using chi-square test. The results of those analyses are presented as numbers and percentages. SPSS Statistics version 16.0 (SPSS, Inc., Chicago, IL, USA) was used to perform all data analyses, and a *p*-value less than 0.05 was considered statistically significant for all tests.

## 3. Results

Among the 1068 samples that were collected from pigs (*n* = 269), chickens (*n* = 318), wild rodents (*n* = 196), pig farmers (*n* = 69), chicken farmers (*n* = 155), and non-farmers (*n* = 61), there were a total of 875 *E. coli* isolates (89.0%) and 108 *Klebsiella* species isolates (11.0%). The prevalence of ceftriaxone, colistin, and meropenem resistance in *E. coli* and *Klebsiella* species isolated from the samples collected from each of the six different sources is shown in Table 1. The overall prevalence of ceftriaxone resistance in *E. coli* and *Klebsiella* species in all samples collected from study animals and humans was significantly higher than that of colistin resistance (32.0% vs. 22.4%, respectively; *p* < 0.01). The overall prevalence of ceftriaxone-resistant and colistin-resistant *E. coli* and *Klebsiella* species isolated from the samples collected from all included animals was 28.4% and 19.3%, respectively (*p* < 0.01). The overall prevalence of ceftriaxone-resistant and colistin-resistant *E. coli* and *Klebsiella* species isolated from the samples collected from all included humans was 42.8% and 30.9%, respectively (*p* < 0.01).

The prevalence of ceftriaxone resistance and colistin resistance in *E. coli* and *Klebsiella* species was highest among animals in the samples collected from pigs, and highest among humans in the samples collected from pig farmers. The prevalence of ceftriaxone resistance in *E. coli* and *Klebsiella* species was significantly higher in pigs than in chickens (64.3% vs. 12.9%, respectively; *p* < 0.01) and wild rodents (64.3% vs. 4.1%, respectively; *p* < 0.01). The prevalence of ceftriaxone resistance in *E. coli* and *Klebsiella* species was also significantly higher in chickens than in wild rodents (12.9% vs. 4.1%, respectively; *p* < 0.01). The prevalence of ceftriaxone resistance in *E. coli* and *Klebsiella* species was higher in pig farmers than in chicken farmers (55.1% vs. 38.7%, respectively; *p* = 0.03) and non-farmers (55.1% vs. 36.1%, respectively; *p* = 0.04); however, the prevalence of ceftriaxone resistance in *E. coli* and *Klebsiella* species in chicken farmers was not significantly higher than that in non-farmers (38.7% vs. 36.1%, respectively; *p* = 0.84). The prevalence of colistin resistance in *E. coli* and *Klebsiella* species was significantly higher in pigs than in chickens (41.3% vs. 9.8%, respectively; *p* < 0.01) and wild rodents (41.3% vs. 4.6%, respectively; *p* < 0.01); however, the prevalence of colistin resistance in *E. coli* and *Klebsiella* species in chickens was not significantly higher than that in wild rodents (9.8% vs. 4.6%, respectively; *p* = 0.05). The prevalence of colistin resistance in *E. coli* and *Klebsiella* species in pig farmers was not significantly higher than that in chicken farmers (34.8% vs. 31.6%, respectively; *p* = 0.75) or non-farmers (34.8% vs. 24.6%, respectively; *p* = 0.28). Moreover, the prevalence of colistin resistance in *E. coli* and *Klebsiella* species in chicken farmers was not significantly higher than that in non-farmers (31.6% vs. 24.6%, respectively; *p* = 0.39). No *E. coli* and *Klebsiella* species isolates were resistant to meropenem.

The prevalence of ceftriaxone, colistin, and meropenem resistance in *E. coli* isolated from the samples collected from each of the six different sources is shown in Table 2. The overall prevalence of ceftriaxone resistance in *E. coli* in all samples collected from study animals and humans was significantly higher than that of colistin resistance (30.5% vs. 21.7%, respectively; *p* < 0.01). The overall prevalence of ceftriaxone-resistant and colistin-resistant *E. coli* isolated from the samples collected from all included animals was 27.1% and 18.8%, respectively (*p* < 0.01). The overall prevalence of ceftriaxone-resistant and colistin-resistant *E. coli* isolated from the samples collected from all included humans was 40.0% and 29.8%, respectively (*p* = 0.01). The prevalence of ceftriaxone resistance and colistin resistance in *E. coli* was highest among animals in the samples collected from pigs, and highest among humans in the samples collected from pig farmers. The prevalence of ceftriaxone resistance in *E. coli* was significantly higher in pigs than in chickens (62.5% vs. 11.3%, respectively; *p* < 0.01) and wild rodents (62.5% vs. 4.1%, respectively; *p* < 0.01). The prevalence of ceftriaxone resistance in *E. coli* was also significantly higher in chickens than in wild rodents (11.3% vs. 4.1%, respectively; *p* < 0.01). The prevalence of ceftriaxone resistance in *E. coli* was higher in pig farmers than in chicken farmers (53.6% vs. 37.4%, respectively; *p* = 0.03) and non-farmers (53.6% vs. 31.2%, respectively; *p* = 0.02); however, the prevalence of ceftriaxone resistance in *E. coli* in chicken farmers was not significantly higher than that in non-farmers (37.4% vs. 31.2%, respectively; *p* = 0.48). The prevalence of colistin resistance in *E. coli* was significantly higher in pigs than in chickens (40.5% vs. 9.1%, respectively; *p* < 0.01) and wild rodents (40.5% vs. 4.6%, respectively; *p* < 0.01); however, the prevalence of colistin resistance in *E. coli* in chickens was not significantly higher than that in wild rodents (9.1% vs. 4.6%, respectively; *p* = 0.08). The prevalence of colistin resistance in *E. coli* in pig farmers was not significantly higher than that in chicken farmers (33.3% vs. 31.0%, respectively; *p* = 0.84) or non-farmers (33.3% vs. 23.0%, respectively; *p* = 0.27). Moreover, the prevalence of colistin resistance in *E. coli* in chicken farmers was not significantly higher than that in non-farmers (31.0% vs. 23.0%, respectively; *p* = 0.31).

The prevalence of ceftriaxone, colistin, and meropenem resistance in *Klebsiella* species isolated from the samples collected from each of the six different sources is shown in Table 3. Since the numbers of the samples of ceftriaxone-resistant and colistin-resistant *Klebsiella* species isolated from each type of sample were much smaller than those of *E. coli*, the results on comparison of the prevalence of ceftriaxone resistance and colistin resistance in *Klebsiella* species were unreliable. Therefore, no comparison of the prevalence of ceftriaxone resistance and colistin resistance in *Klebsiella* species among the various sources of the samples was undertaken.

## 4. Discussion

What qualifies this research as a *One Health* surveillance of AMR study is the fact that samples were collected from food animals (pigs and chickens), humans (pig farmers, chicken farmers, and non-farmers), and wild rodents (as representatives of and to evaluate the environment). This study focused on two types of bacteria (*E. coli* and *Klebsiella* species) and three targeted antibiotics (ceftriaxone, meropenem, and colistin). *Escherichia coli* and *Klebsiella* species were selected because they are the most common types of bacteria in the family Enterobacterales that are colonized in the gastrointestinal tract of humans and animals, and infections due to *E. coli* and *Klebsiella* species are very common community-acquired infections in humans. Extended-spectrum cephalosporin (ceftriaxone)-resistant (or ESBL-producing) and carbapenem (meropenem)-resistant Enterobacterales were categorized by the WHO in 2017 as a ”critical priority” for antibiotic-resistant bacteria and among the list of “priority pathogens” of antibiotic resistant bacteria that are considered to pose the greatest threat to human health. Colistin resistance was also included in our study due to the emergence of a plasmid-mediated gene (*mcr*-1) that encodes a protein that causes Enterobacterales to become resistant to colistin [10]. This colistin-resistance mechanism was first discovered in China in 2015, and this resistance gene can easily be transmitted among bacteria in animals, humans, and the environment. A growing concern is that *mcr*-1-producing colistin-resistant Enterobacterales have now been reported from many countries around the world [11]. The MICs of the antibiotics targeted in this study were determined to identify the phenotype of AMR because the MIC of an antibiotic against a particular bacterium is considered a standard indicator of antibiotic susceptibility.

Although surveillance of phenotypic antibiotic-resistant bacteria, especially extended-spectrum cephalosporin (ceftriaxone)-resistant or ESBL-producing *E. coli* isolated from healthy people, farmers, patients, food animals, pets, food, water from natural sources, wastewater, sewage, rats, flies, and cockroaches in both community and hospital settings in Thailand, has been previously reported, the samples included in those studies were usually collected from different geographic locations, from different sources, and at different times [12,13,14,15,16,17,18,19,20,21,22,23,24,25,26]. Therefore, the prevalence of phenotypic antibiotic-resistant bacteria from the various sources reported in those studies cannot be directly compared. In response, the current study was conducted to obtain and analyze samples collected from food animals, humans, and wild rodents that reside in the same selected communities during the same period so that the prevalence of phenotypic antibiotic-resistant bacteria from various sources could be directly compared.

Ceftriaxone-resistant *E. coli* and *Klebsiella* species in pigs and chickens were common in this study, but the prevalence of ceftriaxone resistance was lower than the rates reported from a previous study conducted in Thailand [14]. In contrast, the prevalence of colistin resistance observed in our study was higher than the rates reported previously from Thailand [25] and Laos PDR [27]. The observed differences between and among studies may be due to differences in study location and the times of the studies. The ceftriaxone and colistin resistance in pigs and chickens observed in this study was highly likely to be associated with antimicrobial use (AMU) because empty bottles of antibiotics and unused antibiotics including penicillin, amoxicillin, norfloxacin, enrofloxacin, tetracycline, colistin, lincomycin, gentamicin, tiamulin, sulfonamides, and ceftriaxone were observed at many farms during sample collection. This observation was the cause of some alarm since several regulations specific to the use of antimicrobials in food animals were published by the Ministry of Agriculture and Cooperatives in Thailand some years ago, which state that all classes of antimicrobials cannot be used to promote food animal growth, that many classes of anti-bacterial agents cannot be used to control or prevent infection in food animals, and that the use of colistin must be limited to treating infection in food animals and to an alternative regimen that cannot exceed several days. The reasons that must explain the observed misuse of antibiotics on these animal farms likely include low levels of knowledge about antimicrobial use, neutral or negative rather than positive attitudes regarding the appropriate use of antimicrobials, and poor practices in using appropriate antimicrobials by chicken and pig farm owners/managers/workers in several provinces in Thailand [28]. Consumption of antibiotics in food animals was reported to be associated with the emergence of antibiotic resistance in bacteria, especially colonized bacteria in the gastrointestinal tract of food animals [24,29,30]. The more frequent antimicrobial resistance observed in pigs compared to chickens might be explained by the much larger volume of medicated feed produced by feed mills for pigs (1055 tons) compared to that produced for chickens (18 tons) in 2019 in Thailand [31]. Therefore, an in-depth study of AMU on the farms included in the present study to investigate the association between AMU and the observed AMR is necessary. The rate of ceftriaxone resistance in wild rodents in this study was much lower than that found in rats that were trapped in open markets in a province located in the central region of Thailand [23]. The rats in open markets were reported to commonly consume food and to be in contact with sewage contaminated with antibiotic-resistant bacteria [23,26]. In contrast, the wild rodents trapped and analyzed in this study were more likely to consume foods found in rice fields and their natural environment. It can also be safely postulated that the rodents in rural communities live in a cleaner environment that is less contaminated with antibiotic-resistant bacteria.

The average rate of ceftriaxone-resistant *E. coli* and *Klebsiella* species among farmers and non-farmers (42.1%) in the present study was comparable to the rates of fecal carriage of ceftriaxone-resistant Enterobacterales reported from several studies on Thai people [12,13,14,15,22]. The observed higher prevalence of ceftriaxone-resistant *E. coli* and *Klebsiella* species among farmers compared to non-farmers in this study was concordant with findings from previous studies, since being a farmer has been found to be a risk factor for fecal carriage of ceftriaxone-resistant Enterobacterales [14,22]. Non-farmers might have ceftriaxone-resistant *E. coli* and *Klebsiella* species in their gastrointestinal tract, potentially as a result of taking antibiotics, from the consumption of foods contaminated with antibiotic-resistant bacteria and/or antibiotic residues, or from exposure to environments contaminated with antibiotic-resistant bacteria, since the contamination of ceftriaxone-resistant Enterobacterales in many fresh foods and selected community environments in Thailand has been found to be common [23,26]. Farmers might have ceftriaxone-resistant *E. coli* and *Klebsiella* species in their gastrointestinal tract transmitted from food animals in addition to the aforementioned possible sources in non-farmers. The average rate of colistin-resistant *E. coli* and *Klebsiella* species among farmers and non-farmers (30.9%) observed in this study was higher than that reported from a previous study conducted in Thailand [14]. The farmers in the current study likely acquired colistin-resistant bacteria from their food animals because many pigs and some chickens at our study farms also had colistin-resistant *E. coli* and *Klebsiella* species. However, the reasons that explain the presence of colistin-resistant *E. coli* and *Klebsiella* species in non-farmers in communities remain unclear. Although 48% of hospitalized patients who received parenteral colistin developed colonization of colistin-resistant *E. coli* and *Klebsiella* species in their gastrointestinal tract, colistin for systemic use has not been available in these communities for a decade, and the contamination rate of colistin-resistant Enterobacterales in fresh foods and selected community environments was found to be extremely low [23,26]. The phenotypes of the ceftriaxone- and colistin-resistant bacteria isolated from animals and humans reported in this study cannot be used to conclude that the same phenotypes of particular antibiotic-resistant bacteria isolated from animals and humans are linked. Therefore, the genotypes of the ceftriaxone- and colistin-resistant bacteria isolated from animals and humans in these study communities need to be analyzed in molecular studies, such as with whole-genome sequencing or polymerase chain reaction, to identify the mechanisms of resistance to antibiotics and to determine the magnitude of the linkage or the similarity of antibiotic-resistant bacteria isolated from different sources, since evidence has been reported that whole bacteria and mobile genetic elements could be transferred from food animals to humans [32]. However, the facilities to perform such molecular studies are not available for routine surveillance of AMR in these communities. Further studies on antibiotic-resistant bacteria and antibiotic residue contamination in foods and the environment in these study communities should also be conducted.

It is a positive that no meropenem resistance was detected among all strains of *E. coli* and *Klebsiella* species isolated from animals and humans in these study communities. Meropenem is a broad-spectrum parenteral antibiotic in the carbapenem group of drugs, which is used to treat infections caused by multi-drug resistant organisms, such as Gram-negative bacteria in hospitalized patients. Carbapenem-resistant, Gram-negative bacteria are usually isolated from hospitalized patients and from the hospital environment, and the prevalence of carbapenem-resistant, Gram- negative bacteria has been increasing over the past decade in hospital settings in Thailand [17]. Carbapenem-resistant, Gram-negative bacterial infection is very difficult to treat, and the mortality rate is high. Since oral carbapenem is not available in Thailand, it is not used in humans or food animals in communities. This lack of availability and use of carbapenems in communities explains the absence of meropenem resistance in all isolates of the *E. coli* and *Klebsiella* species grown from the samples collected in community settings in this study.

The observed high prevalence of AMR from *One Health* surveillance of AMR phenotypes, especially colistin resistance, in food animals and humans in these study communities is worrisome; however, these data can now be used as baseline data for AMR in these study communities. These findings emphasize the urgent need for continued studies to determine the factors that contribute to colistin resistance, the need to reinforce existing regulations specific to antimicrobial use, and the need to implement more appropriate interventions, such as improving understanding of antimicrobial use among farm personnel to reduce colistin resistance in the community. Moreover, repeated *One Health* surveillance of AMR phenotypes should be periodically performed after the aforementioned measures are effectively implemented to evaluate their effectiveness in decreasing AMR phenotypes in these study communities.

## Figures and Tables

**Table 1 antibiotics-11-00556-t001:** Prevalence of ceftriaxone, colistin, and meropenem resistance in *Escherichia coli* and *Klebsiella* species isolated from samples collected from various animal and human sources.

Source of Samples	Ceftriaxone Resistance ^a^	Colistin Resistance ^b^	Meropenem Resistance ^c^
Pig (*n* = 269)	173 (64.3%)	111 (41.3%)	0 (0.0%)
Chicken (*n* = 318)	41 (12.9%)	31 (9.8%)	0 (0.0%)
Wild rodent (*n* = 196)	8 (4.1%)	9 (4.6%)	0 (0.0%)
Pig farmer (*n* = 69)	38 (55.1%)	24 (34.8%)	0 (0.0%)
Chicken farmer (*n* = 155)	60 (38.7%)	49 (31.6%)	0 (0.0%)
Non-farmer (*n* = 61)	22 (36.1%)	15 (24.6%)	0 (0.0%)
Total (*n* = 1068)	342 (32.0%)	239 (22.4%)	0 (0.0%)

^a^ Minimum inhibitory concentration (MIC) of ceftriaxone ≥ 4 μg/mL; ^b^ MIC of colistin ≥ 4 μg/mL; ^c^ MIC of meropenem ≥ 4 μg/mL.

**Table 2 antibiotics-11-00556-t002:** Prevalence of ceftriaxone, colistin, and meropenem resistance in *Escherichia coli* isolated from samples collected from various animal and human sources.

Source of Samples	Ceftriaxone Resistance ^a^	Colistin Resistance ^b^	Meropenem Resistance ^c^
Pig (*n* = 269)	168 (62.5%)	109 (40.5%)	0 (0.0%)
Chicken (*n* = 318)	36 (11.3%)	29 (9.1%)	0 (0.0%)
Wild rodent (*n* = 196)	8 (4.1%)	9 (4.6%)	0 (0.0%)
Pig farmer (*n* = 69)	37 (53.6%)	23 (33.3%)	0 (0.0%)
Chicken farmer (*n* = 155)	58 (37.4%)	48 (31.0%)	0 (0.0%)
Non-farmer (*n* = 61)	19 (31.2%)	14 (23.0%)	0 (0.0%)
Total (*n* = 1068)	326 (30.5%)	232 (21.7%)	0 (0.0%)

^a^ Minimum inhibitory concentration (MIC) of ceftriaxone ≥ 4 μg/mL; ^b^ MIC of colistin ≥ 4 μg/mL; ^c^ MIC of meropenem ≥ 4 μg/mL.

**Table 3 antibiotics-11-00556-t003:** Prevalence of ceftriaxone, colistin, and meropenem resistance in *Klebsiella* species isolated from samples collected from various animal and human sources.

Source of Samples	Ceftriaxone Resistance ^a^	Colistin Resistance ^b^	Meropenem Resistance ^c^
Pig (*n* = 269)	5 (1.9%)	2 (0.7%)	0 (0.0%)
Chicken (*n* = 318)	5 (1.6%)	2 (0.6%)	0 (0.0%)
Wild rodent (*n* = 196)	0 (0.0%)	0 (0.0%)	0 (0.0%)
Pig farmer (*n* = 69)	1 (1.5%)	1 (1.5%)	0 (0.0%)
Chicken farmer (*n* = 155)	2 (1.3%)	1 (0.6%)	0 (0.0%)
Non-farmer (*n* = 61)	3 (4.9%)	1 (1.6%)	0 (0.0%)
Total (*n* = 1068)	16 (1.5%)	7 (0.7%)	0 (0.0%)

^a^ Minimum inhibitory concentration (MIC) of ceftriaxone ≥ 4 μg/mL; ^b^ MIC of colistin ≥ 4 μg/mL; ^c^ MIC of meropenem ≥ 4 μg/mL.

## Data Availability

The study dataset used in this study is available from the corresponding author upon reasonable request.

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
