# Peer review of "One Health* Surveillance of Antimicrobial Resistance Phenotypes in Selected Communities in Thailand"

_antibiotics, 2022, doi:10.3390/antibiotics11050556_

Round 1
Reviewer 1 Report
Comments and suggestions are presented with the attached pdf.

Author Response
Reviewer 1 Comments
antibiotics-1687928: One Health Surveillance of Antimicrobial Resistance Phenotypes in Selected Communities in Thailand
Comments to Authors This study examined the prevalence of ceftriaxone-, colistin-, and meropenem-resistant Escherichia coli and Klebsiella spp. isolates from humans, pigs, chickens, and wild rodents in communities in Northern Thailand. Findings suggest the need for continued monitoring of resistant bacteria from farm and community sources and to include molecular profiling to examine the potential transmission of resistant bacteria between farm animals, farm workers, and non-farm workers in the community.
Suggestions and observations that may improve presentation to readers are presented below.
Additional suggestions regarding wording and presentation are offered in the attached file.
Line 24 were the wild rodents identified as rats or mice only or did the trapped wild rodents include other rodent species? give the rodent species if known.
Response The wild rodents were identified as rats or mice only but we could not tell the rodent species.
78 did the farms have biosecurity programs in place?
Response No, because most of the farms were small domestic farms and these food animals from all farms were not exported.
78 were the farms routinely using antibiotics as grow promoters or therapeutics?
Response Antibiotics were usually used for prevention, control and treatment of infections in these farms.
79 were the pig and chicken farms commercial farms or privately owned?
Response Yes, they were privately owned. We added “privately owned farms” in the revised manuscript.
79 were the pigs of the same or different variety across farms?
Response Yes, the pigs were the same type across farms.
79 were the chickens of the same or different variety across farms?
Response Yes, the chickens were the same type across farms.
83 indicate how the rodents were trapped and sacrificed. were the rodents sacrificed according to an approved animal care and use protocol?
Response The cages with food for trapping rodents were placed at the nearby areas of the farms overnight. The trapped rodents were sacrificed according to an approved animal care and use protocol from the Scientific Research Committee of Kasetsart University, Bangkok, Thailand for animal study (COA no. ACKU 62-VTN-010).
83-88 the numbers of sample sources were taken from Table 1
Response Yes.
94 were the Cary-Blair transport medium tubes prepared in your laboratory or purchased pre-prepared from a commercial source. any either case give supplier name, city, state, province, or district, and country as applicable
Response Cary-Blair transport medium tube was purchased from BOENMED® Boen Healthcare Co., Ltd; Suzhou, China as added in the revised manuscript.
94 how much intestinal content was used as inoculum?
Response The revised manuscript said “Each rectal swab collected from pig and chicken, and intestinal content swab collected from wild rodent was inoculated in 5 ml of tryptic soy broth (TSB) (Sigma-Aldrich Corporation, St. Louis, MO, USA), and it was incubated at 35oC for 16-24 hours or overnight. After incubation, 100 μl of each TSB culture was inoculated onto a MacConkey agar plate supplemented with ceftriaxone (2 μg/ml), and onto a MacConkey agar plate supplemented with colistin (1 μg/ml) to detect antibiotic resistant Gram-negative bacteria.”
98 were samples maintained at room temperature or refrigerated for several days after collection?
Response The rectal swabs from pigs and chickens, and intestinal content swabs collected from wild rodents in Cary-Blair transport medium tubes were kept at room temperature but the stool samples collected from humans were kept in the box that contained ice prior to sending them to the laboratory as described in the revised manuscript.
97 were the manual biochemical tests conducted with a commercial panel test strip? If yes, give the company information. If no, give references and solutions for the biochemical tests.
Response Commercial panel test strip was not used. The revised manuscript said “Suspected colonies of Enterobacterales (lactose fermenter or pink colony) grown on these agar plates were subjected to manual biochemical tests including triple sugar iron test, lysine iron agar slants test, indole test, motility test, ornithine decarboxylase test, urease test, malonate test that were locally prepared in the laboratory using the purchased reagents/materials (Oxoid Ltd; Hampshire, UK or BBL/Difco Diagnostic, Becton Dickinson; MD, USA), and oxidase test (BBL/Difco Diagnostic, Becton Dickinson; MD, USA) to identify E. coli and Klebsiella species.
122 give results separately for E. coli and Klebsiella. give separate results for each district.
Response Table 2 for the results on resistance in E. coli and Table 3 for the results on resistance in Klebsiella spp. along with descriptions of the results on resistance as suggested were added in the result section of the revised manuscript. However, we do not provide separate results for each district because the huge amount of raw data was recorded in Excel file with a lot of codes, abbreviations and data. If we provide more table on separate results for each district, such table will be very complex and very long. Therefore, we would like to refer this suggestion to the data availability statement section that said “The study dataset used in this study is available from the corresponding author upon reasonable request.”
122 if the disclosure of this information is not restricted by confidentiality agreements or government policy, present a map indicating district locations.
Response The responsible authority of study sites for this project asked the researchers for not disclose the names of the districts and the province that were the study sites. Therefore, we were unable to present a map indicating district locations.
146 Table 1 - give results separately in several tables for E. coli and Klebsiella. organize results for each district.
Response Table 2 for the results on resistance in E. coli and Table 3 for the results on resistance in Klebsiella spp. along with descriptions of the results on resistance as suggested were added in the result section of the revised manuscript. However, we do not provide separate results for each district because huge amount of raw data were recorded in Excel file with a lot of codes, abbreviations and data. If we provide more table on separate results for each district, such table will be very complex and very long. Therefore, we would like to refer this suggestion to the data availability statement section that said “The study dataset used in this study is available from the corresponding author upon reasonable request.”
233 something like more cost-effective rep-PCR could be used to compare more isolates than with wgs, including comparisons of antibiotic-resistant and non-antibiotic-resistant bacteria, from the different sources, farms, and districts. dendrograms can be constructed from comparisons of band pattern relatedness.
Response Thank you for your suggestions and I agreed with your comment. However, the manuscript said “…molecular studies, such as whole genome sequencing..”. Therefore, WGS is only an example method of molecular studies. We added “or polymerase chain reaction” right after WGS in the revised manuscript.
364-406 in reference titles, use lowercase font where appropriate.
Response The reference titles were written according to the titles said in the published articles. If we need to change of words of the titles to lower case font, please let me know and we are happy to do so.
Remarks Thank the reviewer very much for editing our manuscript. Nearly all of the reviewer’s suggestions for deleting or adding many words in the comments were made in the revised manuscript as yellow highlights except for the numbers said in 2.2.1. and 2.2.2. that we did not correct them because these numbers were the expected numbers of food animals and humans we wanted to include them into the study. They were not the numbers of the samples we really have collected in the study.

Reviewer 2 Report
I have evaluated the manuscript (Antibiotics-1687928) titled “One Health Surveillance of Antimicrobial Resistance Phenotypes in Selected Communities in Thailand” by Thamlikitkul et. al, discussing surveillance of antimicrobial resistance (AMR) using the One Health approach in selected communities in Thailand. The presentation of results in the manuscript is excellent, and this is a good example of the One Health approach to curb the prevalence of MDR.
All standard methods were used for the experiments and data collection. I found the document interesting for the readers and follow the scope of the journal Antibiotics.
I would recommend the article could be published in Antibiotics after minor corrections. There are technical errors, I hope the editor will take care of them.
The authors need to address the below-mentioned queries.
- Table 1 needs footnotes.
- For Lines 42-44 “The World Health Organization (WHO) in collaboration with the World Organization for Animal Health (OIE) and the Food and Agriculture Organization of the United Nations (FAO) endorsed and launched a global action to combat AMR in 2015” is any recent global action plan to combat AMR available?
- For lines 74-75 “The study was conducted in 5 districts of a province located in the Northern region of Thailand during 2018 and 2019.” Although the studies were completed in 2018 and 2019, however, the author took 2 years to conclude the results. Is any specific reason for the delay?
- For this research work, the weight and age of the pigs and chicken could have been included in supporting information.
- For line 98:” within several days after collection.” The author could mention the range of days instead of citing several days.
- The author could change “ml” to “mL”(Lines 102, 104, 105) throughout the manuscript.
- For Line 103: Change “35 degrees Celsius” to “35 oC”.
- The author could include all MIC results in the supporting information.
- A conclusion is missing.
- The author could include the following relevant references.
(a) McEwen SA, Collignon PJ. Antimicrobial Resistance: a One Health Perspective. Microbiol Spectr. 2018 Mar;6(2). doi: 10.1128/microbiolspec.ARBA-0009-2017. PMID: 29600770.
(b) Thamlikitkul V, Rattanaumpawan P, Boonyasiri A, Pumsuwan V, Judaeng T, Tiengrim S, Paveenkittiporn W, Rojanasthien S, Jaroenpoj S, Issaracharnvanich S. Thailand Antimicrobial Resistance Containment and Prevention Program. J Glob Antimicrob Resist. 2015 Dec;3(4):290-294. doi: 10.1016/j.jgar.2015.09.003. Epub 2015 Oct 14. PMID: 27842876.
Author Response
Response to Reviewer 2
Comments and Suggestions for Authors
I have evaluated the manuscript (Antibiotics-1687928) titled “One Health Surveillance of Antimicrobial Resistance Phenotypes in Selected Communities in Thailand” by Thamlikitkul et. al, discussing surveillance of antimicrobial resistance (AMR) using the One Health approach in selected communities in Thailand. The presentation of results in the manuscript is excellent, and this is a good example of the One Health approach to curb the prevalence of MDR.
All standard methods were used for the experiments and data collection. I found the document interesting for the readers and follow the scope of the journal Antibiotics.
I would recommend the article could be published in Antibiotics after minor corrections. There are technical errors, I hope the editor will take care of them.
Response Thank the reviewer for your compliment.
Reviewer’s comments
The authors need to address the below-mentioned queries.
- Table 1 needs footnotes.
Response The footnotes for Table 1 are added in the revised manuscript.
- For Lines 42-44 “The World Health Organization (WHO) in collaboration with the World Organization for Animal Health (OIE) and the Food and Agriculture Organization of the United Nations (FAO) endorsed and launched a global action to combat AMR in 2015” is any recent global action plan to combat AMR available?
Response There is only one version of global action plan on AMR that has been prepared by the tripartite and launched in 2015.
- For lines 74-75 “The study was conducted in 5 districts of a province located in the Northern region of Thailand during 2018 and 2019.” Although the studies were completed in 2018 and 2019, however, the author took 2 years to conclude the results. Is any specific reason for the delay?
Response This study is also included as the thesis of a PhD student (DS) and she had to attend the classes, do assignments of the PhD course, visit the fields and perform other projects over the past 3 years.
- For this research work, the weight and age of the pigs and chicken could have been included in supporting information.
Response This study included only adult pigs and chickens but the data on weight and age of the study pigs and chickens were not collected. The terms ‘adult pigs’ and ‘adult chickens’ were added in the revised manuscript.
- For line 98:” within several days after collection.” The author could mention the range of days instead of citing several days.
Response The term ‘within 3 days’ was replaced ‘within several days’ in the revised manuscript.
- The author could change “ml” to “mL”(Lines 102, 104, 105) throughout the manuscript.
Response “mL” was changed to “ml” throughout the revised manuscript.
- For Line 103: Change “35 degrees Celsius” to “35 oC”.
Response “35 degrees Celsius” was changed to “35oC” in the revised manuscript.
- The author could include all MIC results in the supporting information.
Response The raw data on MICs of all isolates were recorded in Excel file with a lot of codes and data. Therefore, we would like to refer this suggestion to the data availability statement section that said “The study dataset used in this study is available from the corresponding author upon reasonable request.”
- A conclusion is missing.
Response The instruction for authors said that a section on ‘conclusion’ is optional.
- The author could include the following relevant references.
(a) McEwen SA, Collignon PJ. Antimicrobial Resistance: a One Health Perspective. Microbiol Spectr. 2018 Mar;6(2). doi: 10.1128/microbiolspec.ARBA-0009-2017. PMID: 29600770.
(b) Thamlikitkul V, Rattanaumpawan P, Boonyasiri A, Pumsuwan V, Judaeng T, Tiengrim S, Paveenkittiporn W, Rojanasthien S, Jaroenpoj S, Issaracharnvanich S. Thailand Antimicrobial Resistance Containment and Prevention Program. J Glob Antimicrob Resist. 2015 Dec;3(4):290-294. doi: 10.1016/j.jgar.2015.09.003. Epub 2015 Oct 14. PMID: 27842876.
Response Thank you for the suggestion. However, both articles (one of them is mine) are not research articles and no relevant quantitative results that are related to this study were mentioned. Therefore, I apologize that both suggested articles are not included in the revised manuscript.
